METHODS

# A fast and robust Bayesian nonparametric method for prediction of complex traits using summary statistics

**Geyu Zhou[1], Hongyu Zhao**[1,2]*

**1** Program of Computational Biology and Bioinformatics, Yale University, New Haven, Connecticut, United States of America, **2** Department of Biostatistics, Yale School of Public Health, New Haven, Connecticut, United States of America

* hongyu.zhao@yale.edu

**Data Availability Statement:** Genotype and phenotype data are third party data from UK Biobank (www.ukbiobank.ac.uk) and cannot be shared publicly because only approved users can have access. However, if the user has access to UK

## Abstract

Genetic prediction of complex traits has great promise for disease prevention, monitoring, and treatment. The development of accurate risk prediction models is hindered by the wide diversity of genetic architecture across different traits, limited access to individual level data for training and parameter tuning, and the demand for computational resources. To overcome the limitations of the most existing methods that make explicit assumptions on the underlying genetic architecture and need a separate validation data set for parameter tuning, we develop a summary statistics-based nonparametric method that does not rely on validation datasets to tune parameters. In our implementation, we refine the commonly used likelihood assumption to deal with the discrepancy between summary statistics and external reference panel. We also leverage the block structure of the reference linkage disequilibrium matrix for implementation of a parallel algorithm. Through simulations and applications to twelve traits, we show that our method is adaptive to different genetic architectures, statistically robust, and computationally efficient. Our method is available at https://github.com/eldronzhou/SDPR.

## Author summary

Recently there has been much interest in predicting an individual's phenotype from genetic information, which has great promise for disease prevention, monitoring, and treatment. It has been found that there is great variation in the genetic architecture underlying different complex traits, including the number of genetic variants involved and the distribution of the effect sizes of genetic variants. How to model such genetic contribution is a key aspect for accurate prediction of complex traits. So far, most existing methods make specific assumptions about the shape of the genetic contribution. If these assumptions are not correct, the prediction accuracy might be compromised. Here we propose a method that learns the shape of the genetic contribution without making any explicit assumptions. We found that our method achieved robust performance when compared with other recently developed methods through simulation and real data analysis. Our method is also practically more feasible, since it supports the use of public summary statistics and consumes only small amount of computational resources.

Biobank, we have provided all scripts to reproduce the results in our manuscript on https://github.com/eldronzhou/SDPR_paper. Access to data was obtained under application number 29900. More specifically, in our analysis, we used version 2 of UK Biobank imputed genotype (Identifier: ukb_imp_chr[1-22]_v2.bgen), phenotype (Identifier: ukb29401.enc_ukb) and in hospital records (Identifier: ukb_hesin_diag10.tsv, ukb_hesin_diag9.tsv, ukb_hesin.tsv). For height and BMI, the GWAS summary statistics were downloaded from https://portals.broadinstitute.org/collaboration/giant/images/0/01/GIANT_HEIGHT_Wood_et_al_2014_publicrelease_HapMapCeuFreq.txt.gz and https://portals.broadinstitute.org/collaboration/giant/images/1/15/SNP_gwas_ mc_merge_nogc.tbl.uniq.gz. For HDL, LDL, TC and TG, summary statistics were downloaded from http://csg.sph.umich.edu/willer/public/lipids2013/. For CAD, summary statistics were downloaded from http://www.cardiogramplusc4d.org/data-downloads/. For breast cancer, summary statistics were downloaded from http://bcac.ccge.medschl.cam.ac.uk/bcacdata/oncoarray/oncoarray-and-combined -summary-result/gwas-summary-results-breast-cancer-risk-2017/. For IBD, summary statistics were downloaded from ftp://ftp.sanger.ac.uk/pub/consortia/ibdgenetics/iibdgc-trans-ancestry-filte red-summary-stats.tgz. For T2D, summary statistics were downloaded from https://www.diagram-consortium.org/downloads.html. For SCZ and BP, summary statistics were downloaded from https://figshare.com/articles/dataset/cdg2018-bip-scz/14672019?file=28169349 and https://figshare.com/articles/dataset/cdg2018-bip-scz/14672019?file=28169361.

**Funding:** This work was supported in part by NIH grant NIH GM 134005 and NSF grants DMS 1713120 and 1902903 (H.Z.). The funders had no role in study design, data collection and analysis, decision to publish, or preparation of the manuscript.

**Competing interests:** The authors have declared that no competing interests exist.

## Introduction

Results from large-scale genome-wide association studies (GWAS) offer valuable information to predict personal traits based on genetic markers through polygenic risk scores (PRS) calculated from different methods. For one individual, PRS is typically calculated as the linear sum of the number of the risk alleles weighted by the effect size for each marker, such as single nucleotide polymorphism (SNP) [1]. PRS has gained great interest recently due to its demonstrated ability to identify individuals with higher disease risk for more effective prevention and monitoring [2].

Appropriate construction of PRS requires the development of statistical methods to jointly estimate the effect sizes of all genetic markers in an accurate and efficient way. Statistical challenges associated with the design of PRS methods largely reside in how to account for linkage disequilibrium (LD) among the markers and how to capture the genetic architecture of traits. Meanwhile, practical issues to be addressed include making use of summary statistics as input, as well as reducing the computational burden.

One simple method to compute PRS is to use a subset of SNPs in GWAS summary statistics formed by pruning out SNPs in LD and selecting those below a p value threshold (P+T) [1]. P+T is computationally efficient, though the prediction accuracy can usually be improved by using more sophisticated methods [3]. At present, most of the existing methods that allow the use of summary statistics as input assume a prior distribution on the effect sizes of the SNPs in the genome and fit the model under the Bayesian framework. Methods differ in the choice of the prior distribution. For example, LDpred and LDpred2 assume a point-normal mixture distribution or a single normal distribution [3,4]. SBayesR assumes a mixture of three normal distributions with a point mass at zero [5]. PRS-CS proposes a conceptually different class of continuous shrinkage priors [6]. In reality, there is wide diversity in the distribution of effect sizes for complex traits [7]. Therefore, there may be model specification for choosing a specific parametric prior if the true genetic architecture cannot be captured by the assumed parametric distribution. A natural solution is to consider a generalizable nonparametric prior, such as the Dirichlet process [8]. Dirichlet process regression (DPR) was shown to be adaptive to different parametric assumptions and could achieve robust performance when applied to different traits [9]. However, DPR requires access to individual-level genotype and phenotype data and has expensive computational cost when applied to large-scale GWAS data.

In this work, we derive a summary statistics-based method, called SDPR, which does not rely on specific parametric assumptions on the effect size distribution. SDPR connects the marginal coefficients in summary statistics with true effect sizes through Bayesian multiple Dirichlet process regression. We utilize the concept of approximately independent LD blocks and overparametrization to develop a parallel and fast-mixing Markov Chain Monte Carlo (MCMC) algorithm [10,11]. Through simulations and real data applications, we demonstrate the advantages of our methods in terms of improved computational efficiency and more robust performance in prediction without the need of using a validation dataset to select tuning parameters.

## Methods

### Overview of SDPR

Suppose GWAS summary statistics are derived based on $N$ individuals and $p$ genetic markers, the phenotypes and genotypes can be related through a multivariate linear model,

$$y = X\beta + \epsilon \tag{1}$$

where $y$ is an $N \times 1$ vector of phenotypes, $X$ is an $N \times p$ matrix of genotypes, and $\beta$ is an $p \times 1$ vector of effect sizes. We further assume, without loss of generality, that both $y$ and columns of $X$ have been standardized. GWAS summary statistics usually contain the per SNP effect size $\hat{\beta}$ directly obtained or well approximated through the marginal regression $\hat{\beta} = \frac{X^T y}{N}$. Under the assumption that individual SNP explains relatively small percentage of phenotypic variance, the residual variance can be set to 1 and the likelihood function can be evaluated from

$$\hat{\beta}|\beta \sim N\left(R\beta, \frac{R}{N}\right) \qquad (2)$$

where $R$ is the reference LD matrix [3,12].

Like many Bayesian methods, we assume that the effect size of the i$^{\text{th}}$ SNP $\beta_i$, follows a normal distribution with mean 0 and variance $\sigma^2$. In contrast to methods assuming one particular parametric distribution, we consider placing a Dirichlet process prior on $\sigma^2$, i.e.h a multivariate linear model,

$$\beta_i \sim N(0, \sigma^2), \sigma^2 \sim DP(H, \alpha) \qquad (3)$$

where $H$ is the base distribution and $\alpha$ is the concentration parameter controlling the shrinkage of the distribution on $\sigma^2$ toward $H$. To improve the mixing of MCMC and avoid the informativeness issue of inverse gamma distribution, we follow Gelman's advice to overparametrize the model by writing $\beta_i = \eta \gamma_i$ and use the square of uniform distribution as the base distribution $H$ [13]. This is explained thoroughly in the section Dirichlet Process Prior of S1 Text.

Dirichlet process has several equivalent probabilistic representations, of which stick-breaking process is commonly used for its convenience of model fitting [14]. The stick-breaking representation views the Dirichlet process as the infinite Gaussian mixture model

$$\beta_i \sim \sum_{k=1}^{\infty} p_k N(0, \sigma_k^2), p_k = V_k \prod_{m=1}^{k-1}(1 - V_m), V_k \sim Beta(1, \alpha), \sigma_k^2 \sim H \qquad (4)$$

In practice, truncation needs to be applied so that the maximum number of components of the mixture model is finite. We found that setting the maximum components to 1000 was sufficient for our simulation and real data application because the number of non-trivial components, to which SNPs were assigned, was way fewer than 1000. The first component of the mixture model is further fixed to 0 in analogous to Bayesian variable selection. We designed a parallel MCMC algorithm and implemented it in C++. The details of the algorithm can be found in the section MCMC Algorithm of S1 Text.

## Robust design of the likelihood function

Unlike individual-level data based methods, summary statistics based methods typically rely on external reference panel to estimate the LD matrix $R$. Ideally, the same set of individuals in the reference panel should be used to generate the summary statistics. However, due to the limited access to the individual level data of original GWAS studies, an external database with matched ancestry like the 1000 Genomes Project [15] or UK Biobank [16] is usually used instead to compute the reference LD matrix. It is possible that effect sizes of SNPs in summary statistics deviate from what are expected given the likelihood function and reference LD matrix, especially for SNPs in strong LD that are genotyped on different individuals (Table A in S1 Text). This issue was also noted in the section 5.5 of the RSS paper [12]. Failure to account for such discrepancy can cause severe model misspecification problems for SDPR and possibly other methods. One can derive that, if SNPs are genotyped on different individuals,

then the likelihood function (2) should be modified as

$$\hat{\beta}|\beta \sim N(R\beta, R \circ H) \tag{5}$$

where $\circ$ is the Hadamard product, $H_{ii} = \frac{1}{N_i}$, $H_{ij} = \frac{N_{s,ij}}{N_i N_j}(i \neq j)$, $N_i$ is the sample size of SNP i, $N_j$ is the sample size of SNP j, and $N_{s,ij}$ is the number of shared individuals genotyped for SNPs i and j. Evaluation of likelihood function (5) requires the knowledge about the sample size and inclusion of each study for each SNP. For example, SNPs of GWAS summary statistics of lipid traits were genotyped on two arrays in two separate cohorts (GWAS chip: $N_1 \approx 95{,}000$; Metabochip: $N_2 \approx 94{,}000$) [17]. Based on this information, $N_{s,ij}$ is set to 0 if SNPs i and j were genotyped on different arrays, $N_1$ if SNP i was genotyped on GWAS chip and SNP j was genotyped on both arrays, and $N_2$ if SNP i was genotyped on Metabochip and SNP j was genotyped on both arrays.

In reality, GWAS summary statistics are often obtained through meta-analysis, and information above is generally not available. Besides, double genomic control is applied to many summary statistics, which may lead to deflation of effect sizes [18,19]. Therefore, we consider evaluating the likelihood function from the following distribution.

$$\frac{\hat{\beta}}{c}|\beta \sim N\left(R\beta, \frac{R + NaI}{N}\right) \tag{6}$$

More specifically, the input is divided by a constant provided by SumHer if application of double genomic control significantly deflates the effect sizes [18]. Compared with Eq (5), the correlation between two SNPs is $\frac{R_{ij}}{1+Na}$ rather than $\frac{R_{ij}N_{s,ij}}{\sqrt{N_i N_j}}$. The connection between Eq (6) and LDSC is discussed in the relevant section of S1 Text. For simulated data, $c$ was set to 1 and $\alpha$ was set to 0 for Scenarios 1A-1C, 4 and 5, since there was no above-mentioned discrepancy in these scenarios. In real data application, $Na$ was set to 0.1 except for lipid traits, and $c$ was set to 1 except for BMI (BMI $c = 0.74$ given by SumHer).

## Construction and partition of reference LD matrix

We use an empirical Bayes shrinkage estimator to construct the LD matrix since the external reference panel like 1000G contains a limited number of individuals [20]. LD matrix can be divided into small "independent" blocks to allow for efficient update of posterior effect sizes using the blocked Gibbs sampler [6]. At present, ldetect is widely used for performing such tasks [10]. However, ldetect sometimes produces false positive partitions that violate the likelihood assumption of Eq (2) (Fig A in S1 Text). To address this issue, we designed a simple and fast algorithm for partitioning independent blocks. The new algorithm ensures that each SNP in one LD block does not have nonignorable correlation ($r^2 > 0.1$) with SNPs in other blocks so that the likelihood assumption of Eq (2) is less likely to be violated (Fig A in S1 Text).

## Other methods

We compared the performance of SDPR with seven other methods: (1) PRS-CS as implemented in the PRS-CS software; (2) SBayesR as implemented in the GCTB software (version 2.02); (3) LDpred as implemented in the LDpred software (version 1.0.6); (4) P+T as implemented in the PLINK software (version 1.90) [21]; (5) LDpred2 as implemented in the bigsnpr package (version 1.6.1); (6) Lassosum as implemented in the lassosum package (version 0.4.5) [22]; and (7) DBSLMM as implemented in the DBSLMM package (version 0.21) [23]. We used the default parameter setting for all methods. For PRS-CS, the global shrinkage parameter was

specified as {1e-6, 1e-4, 1e-2, 1, auto}. For SBayesR, gamma was specified as {0, 0.01, 0.1, 1} and pi was specified as {0.95, 0.02, 0.02, 0.01}. For LDpred, the polygenicity parameter was specified as {1e-5, 3e-5, 1e-4, 3e-4, 0.001, 0.003, 0.01, 0.03, 0.1, 0.3, 1, LDpred-Inf}. For P+T, SNPs in GWAS summary statistics were clumped for $r^2$ iterated over {0.2, 0.4, 0.6, 0.8}, and for p value threshold iterated over {5e-8, 5e-6, 1e-5, 1e-4, 5e-4, 1e-3, 1e-2, 0.04, 0.05, 0.1, 0.2, 0.3, 0.4, 0.5, 0.6, 0.7, 0.8, 0.9, 1}. For LDpred2, we ran LDpred2-inf, LDpred2-auto and LDpred2--grid, and reported the best performance of three options. The grid of hyperparameters was set as non-sparse, p in a sequence of 21 values from $10^{-5}$ to 1 on a log-scale, and $h^2$ within {0.7, 1, 1.4} of $h^2_{LDSC}$. For lassosum, lambda was set in a sequence of 20 values from 0.001 to 0.1 on a log-scale, and s within {0.2, 0.5, 0.9, 1}. For DBSLMM, p value threshold was iterated within {$10^{-5}$, $10^{-6}$, $10^{-7}$, $10^{-8}$}, r2 was iterated within {0.05, 0.1, 0.15, 0.2, 0.25}, and $h^2$ was set as $h^2_{LDSC}$. We tuned the parameters for PRS-CS, LDpred, P+T, LDpred2, lassosum, and DBSLMM using the validation dataset.

## Genome-wide simulations

We used genotypes from UK Biobank to perform simulations. UK Biobank's database contains extensive phenotypic and genotypic data of over 500,000 individuals in the United Kingdom [16]. We selected 276,732 unrelated individuals of European ancestry based on data field 22021 and 22006. A subset of these individuals was randomly selected to form the training, validation and test datasets. Training datasets contained 10,000, 50,000, and 100,000 individuals, while validation and test datasets contained 10,000 individuals. We applied quality control (MAF > 0.05, genotype missing rate < 0.01, INFO > 0.3, pHWE > 1e-5) to select 4,458,556 SNPs from the original ~96 million SNPs. We then intersected these SNPs with 1000G HM3 SNPs (MAF > 0.05) and removed those in the MHC region (Chr6: 28–34 Mb) to form a set of 681,828 SNPs for simulation.

To cover a range of genetic architectures, we simulated effect sizes of SNPs under four scenarios: (1)-(3) $\beta_j \sim \pi N\left(0, \frac{h^2}{M\pi}\right) + (1 - \pi)\delta_0$, where $h^2 = 0.5$, $M = 681828$, $\pi$, equaled $10^{-4}$ (scenario 1A), $10^{-3}$ (scenario 1B) and $10^{-2}$ (scenario 1C); (4) $\beta_j \sim \sum_{i=1}^{3} \pi_i N(0, c_i \sigma^2) + (1 - \sum_{i=1}^{3} \pi_i)\delta_0$ where $c = (1, 0.1, 0.01)$, $\pi = (10^{-4}, 10^{-4}, 10^{-2})$ with $\sigma^2$ calculated so that the total heritability equaled 0.5; (5) $\beta_j \sim N\left(0, \frac{h^2}{M}\right)$. Importantly, scenario 1A-1C satisfied the assumption of LDpred/LDpred2, scenario 5 satisfied the assumption of LDpred-inf/LDpred2-inf, whereas scenario 4 satisfied the assumption of SBayesR. Phenotypes were generated from simulated effect sizes using GCTA-sim, and marginal linear regression was performed on the training data to obtain summary statistics using PLINK2 [24,25]. In each scenario, we performed 10 simulation replicates.

We applied different methods on the training data, and used the 10,000 individuals in the validation dataset to estimate the LD matrix. Parameters for LDpred, P+T, PRS-CS, LDpred2, lassosum, and DBSLMM were also tuned using the validation data. We then evaluated the prediction performance on the test data by computing the square of Pearson correlation of PRS with simulated phenotypes.

## Real data application using public summary statistics and UK biobank data

We obtained public GWAS summary statistics for 12 traits and evaluated the prediction performance of each method using the UK Biobank data. Individuals in GWAS do not overlap with individuals in UK Biobank. For this reason, we did not use the latest summary statistics of height and BMI [26]. To standardize the input summary statistics, we generally followed the guideline of LDHub to perform quality control on the GWAS summary statistics [27]. We removed strand ambiguous (A/T and G/C) SNPs, insertions and deletions (INDELs), SNPs

**Table 1. Summary information about the sample size and SNPs in GWAS summary statistics and UK Biobank datasets.** For binary traits, effective sample size was used $(\frac{4*N_{case}*N_{control}}{N_{case}+N_{control}})$ and the validation datasets consisted of equal numbers of cases and controls. If the summary statistics included sample sizes for individual SNPs, the median of all SNPs passing QC was reported. For binary traits, the number of cases and controls were reported in the parenthesis.

| Trait | GWAS sample size | GWAS ref | 1KG HM3 & UKB & GWAS SNPs | UKB validation Sample size | UKB testing sample size |
|---|---|---|---|---|---|
| Height | 252,230 | [29] | 885,791 | 138,066 | 138,066 |
| BMI | 233,766 | [30] | 886,654 | 137,921 | 137,920 |
| HDL | 94,288 | [17] | 868,645 | 37,774 | 37,774 |
| LDL | 89,866 | [17] | 868,179 | 40,807 | 40,807 |
| Total Cholesterol | 94,571 | [17] | 868,167 | 40,898 | 40,898 |
| Triglycerides | 90,989 | [17] | 86,8243 | 40,858 | 40,857 |
| Coronary artery disease | 61,294 (22,233/64,762) | [31] | 814,337 | 4475/4475 | 4475/258,345 |
| Breast Cancer | 227,688 (122,977/105,974) | [32] | 927,706 | 4539/4539 | 4539/133,649 |
| Inflammatory bowel disease | 32,372 (12,882/21770) | [33] | 918,369 | 1840/1840 | 1839/198,815 |
| Type 2 diabetes | 156,109 (26,676/132,532) | [34] | 974,907 | 7240/7240 | 7239/182,292 |
| Bipolar | 41,606 (20,129/21,524) | [35] | 928,032 | 832/832 | 832/176,069 |
| Schizophrenia | 65,955 (33,426/32541) | [35] | 941,216 | 223/223 | 223/203,471 |

with an effective sample size less than 0.67 times the 90th percentile of sample size. SNPs within the MHC region were removed except for IBD, since MHC region plays an important role in autoimmune diseases. The remaining SNPs were then intersected with 1000G HM3 SNPs provided in the PRS-CS reference panel. Table 1 shows the number of SNPs present in the summary statistics for each trait after performing quality control.

For UK Biobank, we first selected unrelated European individuals as we did in simulations. We then applied quality control (MAF > 0.01, genotype missing rate < 0.05, INFO > 0.8, pHWE > 1e-10) to obtain a total of 1,114,176 HM3 SNPs. UK Biobank participants with six quantitative traits-height, body mass index (BMI), high-density lipoproteins (HDL), low-density lipoproteins (LDL), total cholesterol, and triglycerides-were selected based on relevant data fields (Section selection of phenotypes in S1 Text). Selected participants were randomly assigned to form validation and test datasets, each composing half of the individuals. Cases for each of six diseases-coronary artery diseases, breast cancer, inflammatory bowel disease (IBD), type 2 diabetes, bipolar, and schizophrenia-were selected based on self-reported questionnaire and ICD code in the electronic hospital record (EHR). Controls were selected among participants in the EHR based on certain exclusion criteria (Section selection of phenotypes in S1 Text). Validation dataset consisted of an equal number of cases and controls, the rest of which were assigned to the test dataset (Table 1). Random assignments of individuals to validation and test datasets were repeated for 10 times.

For six quantitative traits, we reported the prediction $R^2$ of PRS (variance explained by PRS) defined as $R^2 = 1 - \frac{SS_1}{SS_0}$, where $SS_0$ is the sum of squares of the residuals of the restricted linear regression model with covariates (an intercept, age, sex, top 10 PCs of the genotype data), and $SS_1$ is the sum of squares of the residuals of the full linear regression model (covariates above and PRS). For six diseases, we reported the AUC of PRS only for better comparison of different methods.

## Results

### Adaptiveness of Dirichlet process prior

Theoretically, Dirichlet process as an infinite Gaussian mixture model is able to approximate any continuous parametric distribution, thus including other published parametric

distributions as special cases [28]. For example, the density of Dirichlet process prior adapts well to the density of normal distribution (LDpred-inf), point normal mixture distribution (LDpred/LDpred2), and three-point normal mixture distribution (SBayesR) (Fig B in S1 Text). Compared with SBayesR, Dirichlet process prior does not constrain the relationship between three non-zero normal variance components. We also explicitly incorporate Bayesian variable selection by setting the first variance component as 0, which is different from PRS-CS. The adaptiveness of Dirichlet process prior potentially makes it more robust to the distribution of effect sizes of real traits.

## Simulations

We first compared the performance and computational time of SDPR with DPR in a small-scale simulation setting using 10,000 individuals and 58,432 SNPs on chromosome 1. The effect sizes were generated under the mixture of Dirichlet delta and three normal distributions with total heritability fixed as 0.3. We fitted DPR model with four components and 5000 MCMC iterations, and SDPR model with the input of summary statistics. The average $R^2$ of DPR was 0.227, and the average $R^2$ of SDPR was 0.204 (Fig C in S1 Text). DPR took about 3.5 hours and consumed 10.4 Gb of memory to finish MCMC, while SDPR took only 10 minutes and used 1.1 Gb of memory. This demonstrated the improved computational efficiency of SDPR over DPR without loss of much prediction accuracy.

We then compared the performance of SDPR with several other summary statistics-based methods via genome-wide simulations across different genetic architectures and training sample sizes. Effect sizes of SNPs were simulated under a point-normal mixture model with increasing number of causal variants, a point-three-normal mixture model satisfying SBayesR's assumption, and a normal model satisfying LDpred-inf's assumption (details in Methods). The heritability was fixed as 0.5 and 10 replicates were performed in each simulation setting. Tuning parameters of PRS-CS, LDpred, P+T, LDpred2, lassosum, and DBSLMM were selected using a validation dataset (N = 10,000). 10,000 individuals in the validation dataset were used to construct the LD matrix. We evaluated the prediction performance on the independent test data (N = 10,000) using the squared Pearson correlation coefficient ($R^2$).

The prediction accuracy of all methods generally increased along the sample size of training data (Fig 1 and Tables B-F in S1 Text). Similarly, all methods performed better when the number of causal variants was small. Since the standard error of the regression coefficient estimator in GWAS summary statistics is roughly reciprocal to the square root of the sample size of the training cohort, the dominance of noise over signal poses significant challenges for accurate estimation of effect sizes when the training sample size or per SNP effect size is small.

SDPR, LDpred2, and SBayesR performed better than other methods in the sparse setting (Fig 1 Scenarios 1A-1C and Tables B-E in S1 Text). Consistent with others' findings, we observed that when the genetic architecture was sparse, the performance of LDpred decreased as the training sample size increased [6]. In contrast, LDpred2 performed significantly better than LDpred. Meanwhile, PRS-CS performed worse when the training sample size was small. In the polygenic setting, SDPR and LDpred-inf/LDpred2-inf performed better than other methods (Fig 1 Scenario 5 and Table F in S1 Text). Overall, SDPR and LDpred2 performed well across a range of simulated sparse and polygenic genetic architectures. LDpred2 is expected to perform well in Scenarios 1A-1C and 5 since it satisfied the assumption of LDpred2/LDpred2-inf. The robust performance of SDPR demonstrates the advantage of using Dirichlet process prior to model the genetic architecture.

It is important to note that while SBayesR and SDPR do not need a validation dataset to tune parameters, they may be more susceptible to heterogeneity and errors in the summary

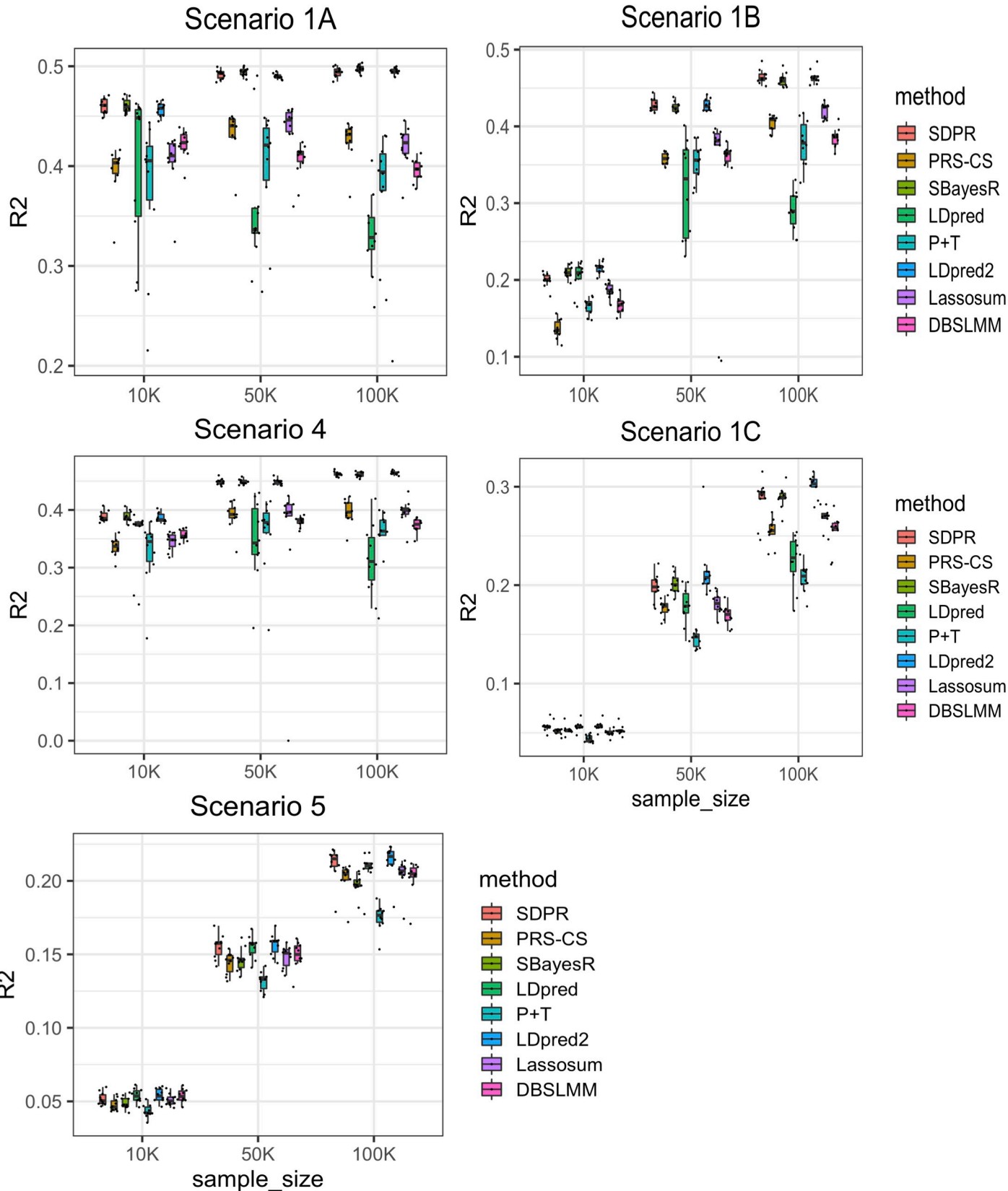

**Fig 1. Prediction performance of different methods on simulated data with varying samples sizes of the training cohort.** Scenarios 1A-1C: mixture of Dirichlet delta and normal distribution (spike and slab) with number of causal SNPs increasing from 100, 1000 to 10000. Scenario 4: mixture of Dirichlet delta and three normal distributions. Scenario 5: single normal distribution. The total heritability in all scenarios was fixed to 0.5. Simulation in each scenario was repeated for 10 times. For each boxplot, the central mark is the median and the lower and upper edges represents the 25th and 75th percentile. The median is recorded in the Table B-E in S1 Text.

statistics. Therefore, we tested whether our modified likelihood function (6) makes SDPR more robust when dealing with discrepancies between summary statistics and reference panel. We generated summary statistics from 50,000 individuals under the same setting as scenario 1B. For half of the SNPs (340,914), linear regression was performed on 40,000 individuals to obtain the marginal effect sizes. According to Eq (5), the correlation of effect sizes of these SNPs would be 80% of what was expected from the reference panel. Such discrepancy indeed caused the divergence of SBayesR, while SDPR with modified likelihood function (6) converged and performed well (N = 50,000, Na = 0.25, $R^2$ = 0.422).

## Real data applications

We compared the performance of SDPR with other methods in real datasets to predict six quantitative traits (height, body mass index, high-density lipoproteins, low-density lipoproteins, total cholesterol, and triglycerides) and six diseases (coronary artery diseases, breast cancer, inflammatory bowel disease, type 2 diabetes, bipolar, and schizophrenia) in UK Biobank. We obtained public GWAS summary statistics of these traits and performed quality control to standardize the input (details in Methods; Table 1). A total of 503 1000G EUR individuals were used to construct the reference LD matrix for SDPR, PRS-CS, LDpred, P+T, LDpred2, lassosum, and DBSLMM. For SBayesR, we used 5000 EUR individuals in UK Biobank to create the LD matrix (shrunken and sparse) instead, as it was reported to have suboptimal prediction accuracy when using 1000G samples [5].

For six continuous traits, the prediction performance was measured by variance of phenotype explained by PRS (Fig 2 and Table G in S1 Text). Overall, SDPR, PRS-CS and LDpred2 performed better than other methods, and there was minimal difference of these three methods. In terms of ranking, SDPR and PRS-CS performed best for height. SDPR and LDpred2 performed best for BMI. SDPR performed best for HDL, LDL and total cholesterol, while PRS-CS performed best for triglycerides. We observed convergence issues when running SBayesR on these traits, and followed its manual to filter SNPs based on GWAS P-values and LD R-squared (—p-value 0.4—rsq 0.9). The filtering approach improved the prediction performance of SBayesR, but it still failed to achieve the top tier performance. We suspect that the convergence issue of SBayesR was also caused by the violation of the likelihood assumption, similar to what we observed in the simulation. To address this issue, our approach of modifying the likelihood function might be better than the simple filtering approach used in SBayesR and P+T as it retained all SNPs for prediction.

For six disease traits, the prediction performance was measured by AUC of PRS only (Fig 3 and Table H in S1 Text). Overall, SDPR achieved top tier performance (within 0.003 difference of AUC of the best method) for five out of six diseases. In terms of ranking, LDpred and LDpred2 performed best for coronary artery disease. SDPR and PRS-CS performed best for breast cancer. LDpred2 performed best for IBD. For schizophrenia and type 2 diabetes, SBayesR performed best. LDpred, SDPR, LDpred2 and SBayesR performed best for bipolar.

Consistent with simulations, SBayesR performed similarly to SDPR when there was no convergence issue (IBD, type 2 diabetes, schizophrenia, bipolar vs height, lipid traits). In general, PRS-CS performed better when the training sample size was large (height and breast cancer vs IBD and type 2 diabetes) and LDpred performed better when the training sample size was

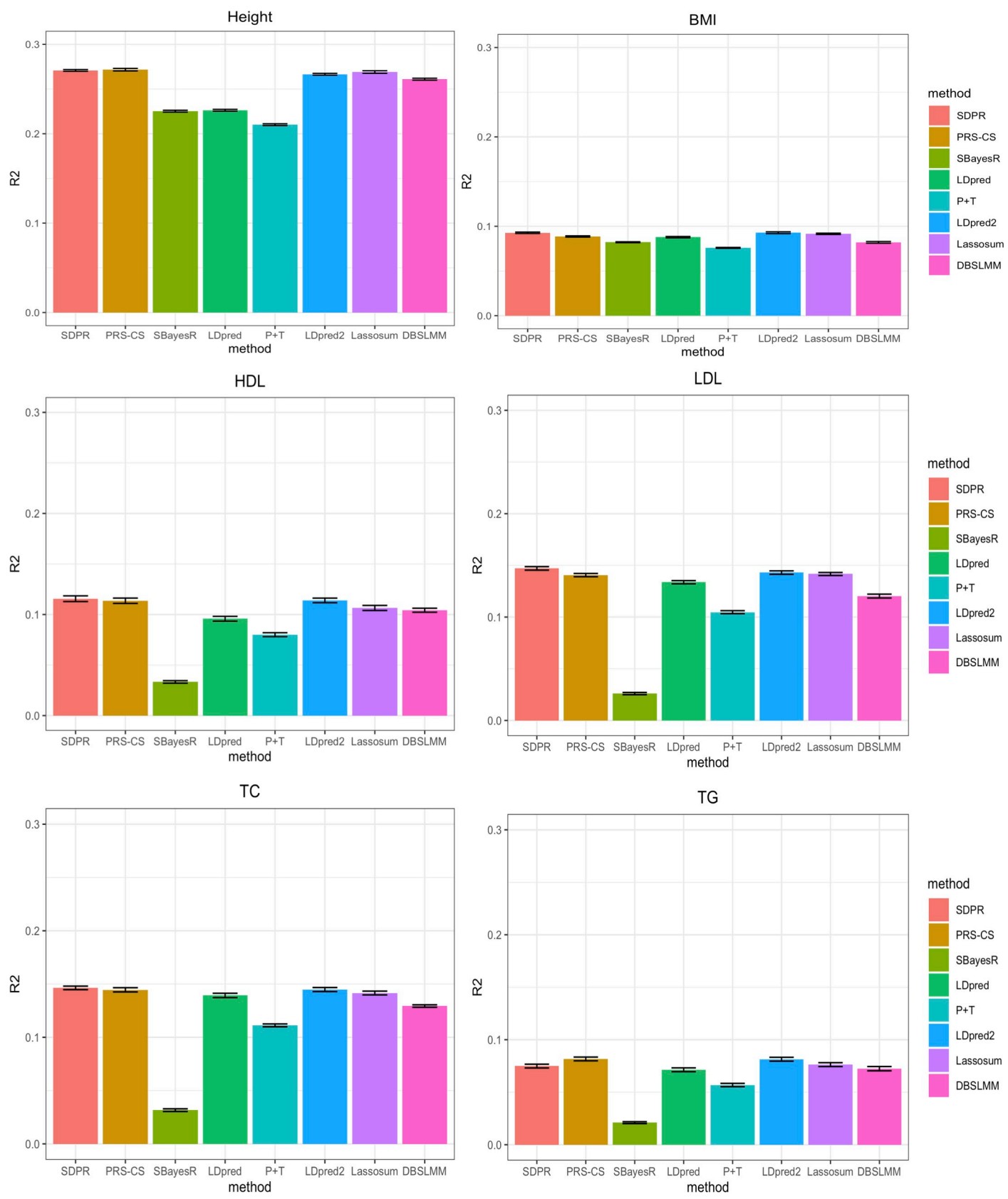

**Fig 2. Prediction performance of different methods for six quantitative traits in the UK Biobank.** Selected participants with corresponding phenotypes were randomly assigned to form validation and test dataset, each composing half of individuals. For PRS-CS, LDpred, P+T, LDpred2, lassosum, and DBSLMM, parameters were tuned based on the performance on the validation dataset. We repeated the split and tuning process 10 times. The mean of variance of phenotypes explained by PRS across 10 random splits was reported in the Table G in S1 Text.

small (coronary artery disease, IBD vs height, breast cancer). LDpred2 performed significantly better than LDpred, achieving highly competitive performance. SDPR performed best among methods (PRS-CS auto, SBayesR, LDpred2 auto) without the need of parameter tuning (Table I and J in S1 Text). Taken together, our design of the likelihood function and usage of Dirichlet process prior empowers SDPR with generally robust performance across different genetic architectures and training sample sizes.

## Computational time

SDPR is implemented in C++ to best utilize the resources of high-performance computing facilities. SDPR optimizes the speed of the computational bottleneck by using SIMD programming, parallelization over independent LD blocks, and high-performance linear algebra library. Besides, SDPR by default runs analysis on each chromosome in parallel because the genetic architecture may be different across chromosomes. We benchmarked the computational time and memory usage of each method on an Intel Xeon Gold 6240 processor (2.60 GHZ). For SDPR and PRS-CS, we paralleled computation over 22 chromosomes and used three threads per chromosome for the linear algebra library ($22 \times 3 = 66$ threads in total). Time and memory usage were reported for the longest chromosome, which was the rate limiting step. For LDpred, SBayesR and P+T, no parallelization was used. LDpred2 was run in the genome-wide mode with 10 threads for parallel computation. DBSLMM and lassosum were run with 3 threads for parallel computation. The evaluation was based on a fixed number of MCMC iterations-1000 for SDPR and PRS-CS (default), 4000 for SBayesR (non-default but achieved generally good performance in simulations and real data application), 100 for LDpred (default), 1000 for LDpred2 (default). One should keep in mind that the number of MCMC iterations and threads for parallel computation affects the computation time significantly, though we did not explore it in this paper since each method also has different convergence and computational properties.

Table 2 shows that SDPR was able to finish the analysis in 15 minutes for most traits and required no more than 3 Gb of memory for each chromosome. SBayesR was also fast but the memory usage was significant for five diseases as no SNPs were removed to improve the convergence. The speed of PRS-CS, LDpred, P+T, LDpred2, lassosum, and DBSLMM was impeded by the need of iterating over tuning parameters. PRS-CS used less memory because the largest size of LD blocks output by ldetect was smaller compared with SDPR.

## Discussion

Building on the success of genome wide association studies, polygenic prediction of complex traits has shown great promise with both public health and clinical relevance. Recently, there is growing interest in developing non-parametric or semi-parametric approaches that make minimal assumptions about the distribution of effect sizes to improve genetic risk prediction [9,36,37]. However, these methods either require access to individual-level data (DPR) [9], external training datasets (NPS) [36], or do no account for LD (So's method) [37]. Other widely used methods usually make specific parametric assumptions, and require external validation or pseudo-validation datasets to optimize the prediction performance [3,6,22]. To address the limitations of the existing methods, we have proposed a non-parametric method

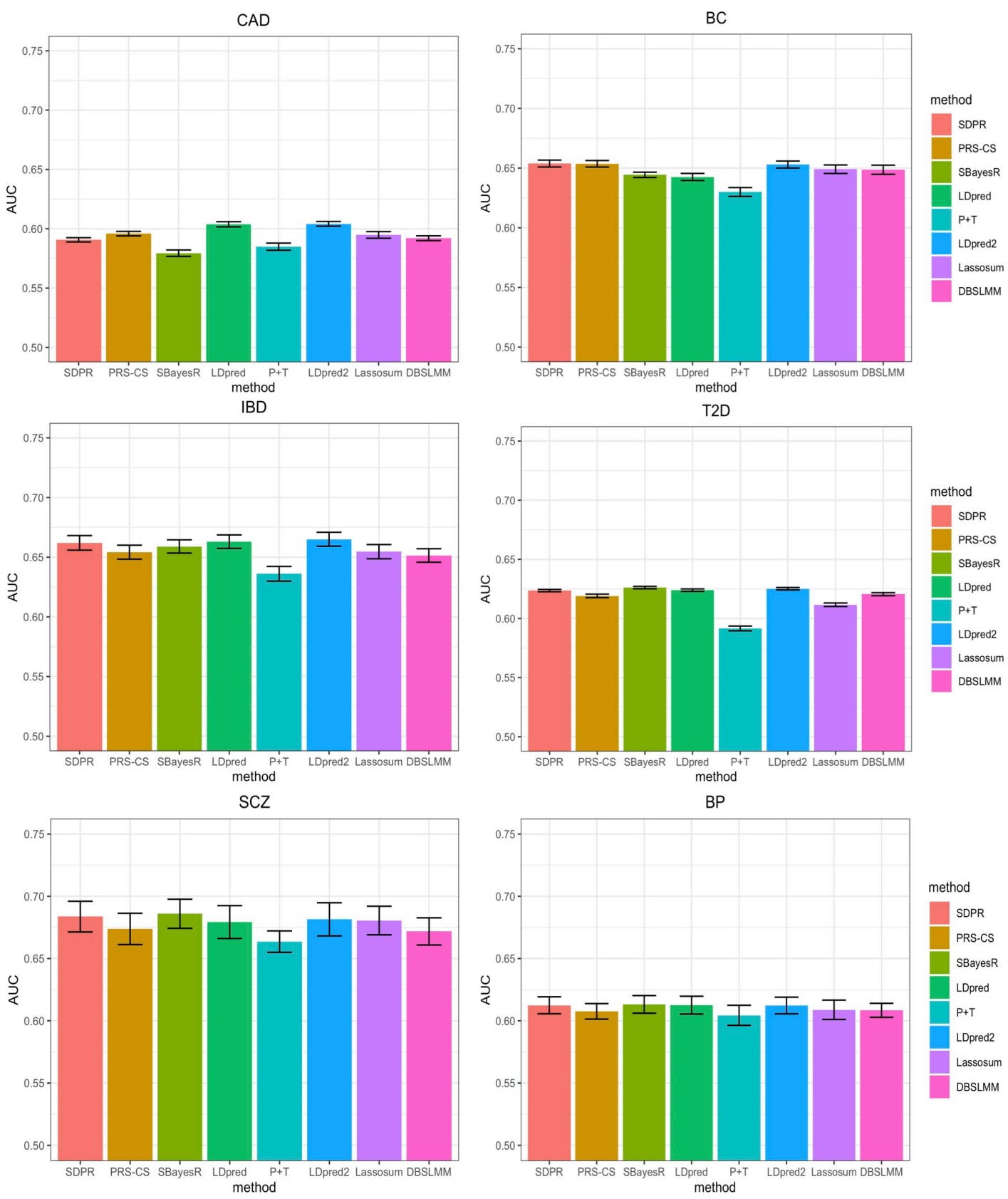

**Fig 3. Prediction performance of different methods for 6 diseases in the UK biobank.** Selected participants with corresponding diseases were randomly assigned to form validation and test dataset (Table 1). For PRS-CS, LDpred, P+T, LDpred2, lassosum and DBSLMM, parameters were tuned based on the performance on the validation dataset. We repeated the split and tuning process for 10 times. The mean AUC across 10 random splits was reported in the Table H in S1 Text.

SDPR that is adaptive to different genetic architectures, statistically robust, and computationally efficient. Through simulations and real data applications, we have illustrated that SDPR is practically simple, fast yet effective to achieve competitive performance.

One of the biggest challenges of summary statistics based method is how to deal with mismatch between summary statistics and reference panel. Based on our experience, misspecification of correlation of marginal effect sizes for SNPs in high LD can sometimes cause severe convergence issues of MCMC, especially for methods not relying on parameter tuning. Our investigation revealed that even when estimating LD from a perfectly matched reference panel, if SNPs were genotyped on different individuals, the correlation/covariance of marginal effect sizes in the summary statistics can be different from what is expected from the reference panel. We proposed a modified likelihood function to deal with this issue and observed improved convergence of MCMC. Our approach may be applied in a broader setting given that many summary statistics based methods assume $\hat{\beta}\beta \sim N\left(R\beta, \frac{R}{N}\right)$ or $z|\beta \sim N(R\sqrt{N}\beta, R)$. When the sample size is small, the noise and heterogeneity of GWAS summary statistics poses more challenge for methods trying to learn every parameter from data (PRS-CS auto, LDpred2-auto, SBayesR, and SDPR). Under such circumstances, it is advantageous for methods like LDpred/LDpred2 to use an independent validation dataset to select the optimal parameters.

Although we have focused on the polygenic prediction of SDPR in this paper, it can provide estimation of heritability, genetic architecture, and posterior inclusion probability (PIP) for fine mapping. These issues will be fully explored in our future studies. SDPR can also be extended as a summary statistics-based tool to predict gene expression level for transcriptome wide association studies since a previous study has shown that individual level data based Dirichlet process model improves transcriptomic data imputation [38].

Although our method has robust performance in comparison with other methods, we caution that currently for most traits the prediction accuracy is still limited for direct application in clinical settings. From our perspective, there are three factors that affect the prediction accuracy. First, how much heritability is explained by common SNPs for diseases and complex

**Table 2. Computational time and memory usage of different methods for 12 traits.** The computational time is in hours. Memory usage of each method, as listed in the parenthesis, is measured in the unit of Gigabytes (Gb). We did not include the time of computing PRS in the validation and test datasets except for P+T, lassosum, LDpred2, and DBSLMM, because such computation was non-trivial for methods with a large grid of tuning parameters.

| Trait | SDPR | PRS-CS | SBayesR | LDpred | P+T | LDpred2 | Lassosum | DBSLMM |
|---|---|---|---|---|---|---|---|---|
| Height | 0.20 (2.4) | 2.5 (0.7) | 0.92 (12.6) | 5.0 (15.5) | 0.6 (1.1) | 5.5 (31.2) | 0.50 (2.6) | 1.7 (1.1) |
| BMI | 0.18 (2.4) | 2.8 (0.7) | 0.50 (7.6) | 4.9 (15.1) | 0.5 (1.1) | 5.4 (30.8) | 0.45 (2.6) | 0.60 (1.1) |
| HDL | 0.20 (2.4) | 1.6 (0.7) | 0.68 (8.5) | 5.1 (15.6) | 0.5 (1.1) | 3.9 (31.7) | 0.41 (2.2) | 0.44 (1.1) |
| LDL | 0.22 (2.4) | 2.2 (0.7) | 0.67 (8.7) | 5.1 (15.6) | 0.6 (1.1) | 5.5 (31.6) | 0.42 (2.2) | 0.61 (1.1) |
| Total cholesterol | 0.25 (2.4) | 2.2 (0.7) | 0.48 (8.7) | 5.1 (15.4) | 0.5 (1.1) | 4.1 (31.7) | 0.40 (2.6) | 0.60 (1.1) |
| Triglycerides | 0.21 (2.4) | 2.2 (0.7) | 0.50 (8.3) | 5.1 (15.5) | 0.5 (1.1) | 3.5 (31.6) | 0.42 (2.6) | 0.62 (1.1) |
| Coronary artery disease | 0.23 (2.3) | 1.9 (0.7) | 0.39 (7.0) | 4.7 (14.0) | 0.3 (1.1) | 3.5 (27.1) | 0.33 (2.2) | 0.77 (1.1) |
| Breast cancer | 0.20 (2.9) | 2.7 (0.7) | 0.63 (42.3) | 5.5 (16.4) | 0.5 (1.1) | 4.6 (37.1) | 0.42 (2.7) | 0.65 (1.1) |
| IBD | 0.28 (2.8) | 2.2 (0.7) | 0.78 (39.5) | 5.1 (16.0) | 0.6 (1.1) | 3.7 (33.4) | 0.45 (2.7) | 0.68 (1.1) |
| Type 2 diabetes | 0.31 (2.9) | 2.4 (0.7) | 0.87 (47.4) | 5.5 (17.4) | 0.5 (1.1) | 4.5 (37.2) | 0.51 (2.8) | 0.63 (1.2) |
| Schizophrenia | 0.28 (2.7) | 2.3 (0.7) | 2.6 (42.1) | 5.3 (16.4) | 0.5 (1.1) | 4.4 (36.8) | 0.43 (2.3) | 0.64 (1.1) |
| Bipolar | 0.28 (2.8) | 2.2 (0.7) | 1.7 (43.8) | 5.3 (16.3) | 0.5 (1.1) | 4.4 (36.8) | 0.45 (2.6) | 0.64 (1.1) |

traits? Second, if diseases or complex traits have relatively moderate heritability, is the GWAS sample size large enough to allow accurate estimation of effect sizes? Third, if the above two conditions are met, is a method able to have good prediction performance? The first two questions have been discussed in the literatures [7,39,40]. As for method development, we have focused on addressing the third question in this paper, and think SDPR represents a solid step in polygenic risk prediction.

Finally, we provide two technical directions for further development of SDPR. First, SDPR may have better performance after incorporating functional annotation as methods utilizing functional annotation generally perform better [41]. Second, studies have shown that PRS developed using EUR GWAS summary statistics does not transfer well to other populations [42,43]. We can further modify the likelihood function to account for different LD patterns across populations to improve the performance of trans-ethnic PRS.

## Supporting information

**S1 Text. Supplementary note to explain methods in details.**
(DOCX)

## Acknowledgments

We conducted the research using the UK Biobank resource under an approved data request (ref: 29900). We sincerely thank GIANT, GLGC, CARDIoGRAMplusC4D, BCAC, IIBDGC, DIAGRAM, and PGC consortia for making their GWAS summary data publicly accessible.

## Author Contributions

**Conceptualization:** Geyu Zhou, Hongyu Zhao.

**Data curation:** Geyu Zhou.

**Formal analysis:** Geyu Zhou.

**Funding acquisition:** Hongyu Zhao.

**Investigation:** Geyu Zhou, Hongyu Zhao.

**Methodology:** Geyu Zhou.

**Software:** Geyu Zhou.

**Supervision:** Hongyu Zhao.

**Writing – original draft:** Geyu Zhou, Hongyu Zhao.

**Writing – review & editing:** Geyu Zhou, Hongyu Zhao.

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
