## [Decision Letter · Decision Letter 0]

14 Mar 2021

Dear Dr Zhao,

Thank you very much for submitting your Methods entitled 'A fast and robust Bayesian nonparametric method for prediction of complex traits using summary statistics' to PLOS Genetics.  We apologise for the slow turnaround, which was largely caused by a difficulty finding suitable reviewers.

Your paper has now been reviewed by three expert reviewers, who provide comments below. While they had positive comments, they also raised important concerns, which mean that the paper can not be accepted in its current state. However, we are willing to consider a revised version, that addresses the comments of the reviewers. In particular, you will note that they have requested additional methods are included in the comparisons, including DPR, LassoSum and LDPred2. Further, they have questions about the robustness of your method, including whether it is necessary to first exclude large effects.

Should you decide to revise the manuscript for further consideration here, your revisions should address all the specific points made by each reviewer. We will also require a detailed list of your responses to the review comments and a description of the changes you have made in the manuscript.

If you decide to revise the manuscript for further consideration at PLOS Genetics, please aim to resubmit within the next 60 days, unless it will take extra time to address the concerns of the reviewers, in which case we would appreciate an expected resubmission date by email to plosgenetics@plos.org.

[LINK]

We are sorry that we cannot be more positive about your manuscript at this stage. Please do not hesitate to contact us if you have any concerns or questions.

Yours sincerely,

Doug Speed

Guest Editor

PLOS Genetics

David Balding

Section Editor: Methods

PLOS Genetics

Reviewer's Responses to Questions

**Comments to the Authors:**

Reviewer #1: Zhou and Zhao present SDPR, a fast and robust Bayesian nonparametric method for building polygenic scores based on summary statistics. Lately, many Bayesian methods have been developed to derive polygenic scores based on summary statistics as well. They all are very similar in the algorithm and implementation they use, with the exception that they assume different prior distributions. At first sight, SDPR does not seem very different.

Therefore there are a few comments/questions I would like the authors to address so that I can better understand what SDPR adds to the current literature. Then and only then could I recommend SDPR to be published in PLOS Genetics.

Major comments (in no particular order):

- I first want to congratulate the authors for sharing their analysis code on GitHub.

- The authors claim that their method is more robust to different architectures because they use Dirichlet processes. Before PRS-CS claimed the same thing by introducing continuous shrinkage priors and SBayesR by using a point-mass at 0 and a mixture of 3 Normal distributions (instead of using only one in e.g. LDpred). The authors present a hierarchical Bayesian framework similar (for non-Bayesian experts) to the one used in PRS-CS and present the Dirichlet process as an infinite Gaussian mixture model. Therefore I wonder how different SDPR is from PRS-CS and SBayesR. I would urge the authors to explain these differences in more detail and how they think this could make SDPR more robust to different architectures than other published methods.

- Sometimes I wonder if allowing too much flexibility in the model is really a good thing? I guess more flexibility can hinder convergence or even make fitting diverge as it is sometimes the case in SBayesR. Is this why LDpred works better when summary statistics come from a GWAS with a small sample size? I guess that SDPR would work only with GWAS summary statistics for large sample size then? The authors could comment on this to expand the Discussion.

- Again, maybe a naive question, but the authors present their method as nonparametric, but I see that many parametric distributions are assumed as priors for everything. What does “nonparametric” mean here? How is SDPR less parametric than e.g. LDpred2-auto which just assumes that beta ~ N(0, h2/(Mp)), p~U(0,1) and estimate h2=beta^T R beta?

- The authors say they “refine the commonly used likelihood assumption to deal with the discrepancy between summary statistics and external reference panel”. In my opinion, it is a very interesting point of this paper. However, the authors use simulation scenarios that do not allow to show this at all. Clear comparisons that show how this change makes SDPR more robust to the discrepancy between summary statistics and external reference panels would be well received. An ultimate test (which might be too difficult) would be to use a reference panel from individuals of e.g. South Asian ancestry, or at least a mixture of European and South Asian ancestry.

- On the same point, is quality control not enough to control the discrepancy between summary statistics and external reference panel? E.g. the ones proposed in LDpred2 (https://doi.org/10.1093/bioinformatics/btaa1029) or in DENTIST (https://doi.org/10.1101/2020.07.09.196535), or maybe even better, using both.

- Again talking about robustness of methods, the authors remove the MHC region, which is usually performed for methods that are not robust to long-range LD regions. Is it the case for SDPR? How much signal is lost when removing this region? SNPs with extremely large effects are also removed, which should be the ones that are most useful for prediction. I wonder then if SDPR is really a robust method. By the way, these two removals of SNPs should never be called “quality control”.

- A simulation scenario with p=0.1 (and even p=1) should be added since many traits are thought to be very polygenic.

- Two methods that have been shown to perform very well in the literature, namely lassosum and LDpred2, are missing from the comparisons.

Minor comments:

- Figures could be made nicer and more readable by using ‘+ theme_bw(16)’ (or even 18 if necessary).

- Maybe report the AUC of the PGS only, not of the full model, for better comparison of methods, and starting the y-axis at 0.5.

- In the introduction, what does “reparameterization” refer to here?

- Is correction for genomic control really necessary if using Z-Scores instead of p-values?

Reviewer #2: The manuscript describes a summary statistics based non-parametric method, SDPR, for genetic prediction of complex traits. The authors applied SDPR through four simulation settings and applications to twelve traits in the UK Biobank. In the simulations, SDPR works quite similarly as SBayesR but outperforms the other methods. In the UK Biobank applications, SDPR outperforms the other methods in more than half of the examined traits. Overall, I think this is a very nice method that adapts DPR to summary statistics and large-scale data applications. It is a timely contribution to the PRS field. The developed method has the potential to be widely used. I only have a few main comments:

1. In our experience, the four compared methods (PRS-CS, SBayesR, LDpred, P+T) are not among the most accurate PRS methods and are usually quite easy to outperform. It would be useful to add comparisons with lassosum and DBSLMM, both of which work quite well across a range of settings.

2. As a related note, I am a bit surprised to see that SDPR performs similarly as SBayesR in all simulation settings. It would be helpful to identify some simulation settings where SDPR can clearly outperform SBayesR. SDPR is a polygenic model that assumes all SNPs to have non-zero effects, while SBayesR is a sparse model that sets a proportion of SNPs to have zero effects. Therefore, it might be helpful explore a few polygenic settings where all or a large fraction of SNPs have non-zero effects. This way, the new simulation results will become well aligned with the current real data results.

3. Given that SDPR is a summary statistics version of DPR, it would be beneficial to compare SDPR with DPR in some small-scale simulations. These simulations can help benchmark the computational gains brought by SDPR over DPR and evaluate the potential accuracy loss in SDPR as compared to DPR.

Minor comment:

1. On line 56-357 on page 20, the authors mentioned that "However, these methods either require external training datasets or do no account for LD". This statement on the ref 9 (DPR) does not appear to be accurate. As far as I am aware of, DRP does not require an external training dataset and does account for LD; it just does not model summary statistics as SDPR does.

Reviewer #3: The authors present a non-parametric PRS based on a new idea for data-driven adaptive modeling of the underlying effect size distribution. Extending on the previous work on Dirichlet process regression, which required individual-level genotype data, they propose a new MCMC algorithm that allows for the training of a PRS with summary-level GWAS data and reference LD panel. The performance of the new approach is benchmarked in simulated and real data. The work is solid and will be of great interest to PLoS Genetics readership if the following concerns are addressed.

Simulation scenarios: I am somewhat disappointed that the new method does not outperform SBayesR in simulation. Does SDPR outperform SBayesR when the effect sizes are sampled from a distribution that is very different from the BayesR model? SDPR performs very robustly in real data; however, it's difficult to know whether this robustness comes from the flexible non-parametric prior or some other features of the algorithm.

PRS-CS: I think PRS-CS authors share codes to prepare custom LD matrices upon request. Comparisons to PRS-CS are particularly important here because it is more similar to SDPR than SBayesR is. I agree that the Dirichlet process would be more adaptive and flexible than PRS-CS's Strawderman-Berger prior on \\sigma^2, but it needs to be shown clearly in which conditions this leads to higher accuracy.

Extremely large effects: the authors "excluded extremely large effects (z2 > 80) to improve the convergence of MCMC" (p. 10) along with MHC. I think this decision is understandable and practical but wonder if SDPR is less accurate in handling large effect tails compared to other methods.

**Have all data underlying the figures and results presented in the manuscript been provided?**

Reviewer #1: Yes

Reviewer #2: Yes

Reviewer #3: Yes

PLOS authors have the option to publish the peer review history of their article (what does this mean?). If published, this will include your full peer review and any attached files.

Reviewer #1: No

Reviewer #2: No

Reviewer #3: No

---

## [Decision Letter · Decision Letter 1]

24 Jun 2021

Dear Dr Zhao,

Thank you very much for submitting a revised version of your Methods entitled 'A fast and robust Bayesian nonparametric method for prediction of complex traits using summary statistics' to PLOS Genetics.

Your article has now been reviewed by the original three reviewers. You will see from their comments that they are all satisfied that you addressed their original comments. However Reviewer 3 has a question about your Github page and version consistency. Therefore, please can you address this new comment of Reviewer 3 (as well as correct the typos noted by Reviewer 1).

Specifically, we ask that you:

[LINK]

Yours sincerely,

Doug Speed

Guest Editor

PLOS Genetics

David Balding

Section Editor: Methods

PLOS Genetics

Reviewer's Responses to Questions

**Comments to the Authors:**

Reviewer #1: I thank the authors for their revised work and do not have any further comment.

Just two typos:

- Hammond -> Hadamard

- ldtect -> ldetect

Reviewer #2: All my comments are well addressed and the paper is ready to publish. I just want to bring up one minor technical inconsistency between the github code and the main text. On line 177, the authors listed a set of hyper-parameter choices that were used for fitting DBSLMM, which implies the tuning version of DBSLMM was used. However, based on the github code, it seems that the automatic version of DBSLMM was used which estimates all hyper-parameters based on the training data automatically (since no validation data was used in the fitted code there). The automatic version has slightly worse performance than the tuning version, but comparing to either version is completely fine and would demonstrate the benefits of SDPR. I don't know if I went to the wrong github site; if not, it would be important to keep the main text consistent with the github code.

Reviewer #3: The authors addressed all concerns.

**Have all data underlying the figures and results presented in the manuscript been provided?**

Reviewer #1: Yes

Reviewer #2: None

Reviewer #3: Yes

PLOS authors have the option to publish the peer review history of their article (what does this mean?). If published, this will include your full peer review and any attached files.

Reviewer #1: No

Reviewer #2: No

Reviewer #3: No

---

## [Editor Report · Decision Letter 2]

5 Jul 2021

Dear Dr Zhao,

Thank you for making the requested changes to your article "A fast and robust Bayesian nonparametric method for prediction of complex traits using summary statistics", and for explaining the GitHub concerns. I am happy to say your article has been editorially accepted for publication in PLOS Genetics. Congratulations!

Yours sincerely,

Doug Speed

Guest Editor

PLOS Genetics

David Balding

Section Editor: Methods

PLOS Genetics

Comments from the reviewers (if applicable):

**Data Deposition**

http://datadryad.org/submit?journalID=pgenetics&manu=PGENETICS-D-20-01887R2

**Press Queries**

---

## [Editor Report · Acceptance letter]

20 Jul 2021

PGENETICS-D-20-01887R2 

A fast and robust Bayesian nonparametric method for prediction of complex traits using summary statistics 

Dear Dr Zhao, 

We are pleased to inform you that your manuscript entitled "A fast and robust Bayesian nonparametric method for prediction of complex traits using summary statistics" has been formally accepted for publication in PLOS Genetics! Your manuscript is now with our production department and you will be notified of the publication date in due course.

With kind regards,

Agota Szep

PLOS Genetics

On behalf of:
